# Analysis of Lateral Displacement of Pile Foundation Caused by Large-Diameter Shield Tunneling

**Yang Sun [1,2,*], Faxin Wang [1], Zoulei Meng [1] and Chongxiao Wang [1]**

1    College of Harbour, Coastal and Offshore Engineering, Hohai University, Nanjing 210098, China; faxin.wang@hhu.edu.cn (F.W.); mengzoulei@hhu.edu.cn (Z.M.); 191303020013@hhu.edu.cn (C.W.)
2    Huai'an Research Institute, Hohai University, Huai'an 223005, China
*    Correspondence: sunyang_hhu@hhu.edu.cn; Tel.: +86-25-8378-6619

**Abstract:** The aim of this research is to study the lateral deformation characteristics of the pile foundation in front of a shield construction and accurately predict its displacement in the shield tunneling direction. The impact of deep-shield construction on the pile foundations was analyzed using the Mindlin elastic solution to determine the lateral displacement. The Kerr foundation model and other factors, like additional shield thrust and uneven shell friction, were considered. The study assessed the impact of the incision distance, shield outer diameter, and additional thrust on pile displacement. The theoretical and numerical solutions of lateral displacement at various shield construction stages were compared to determine the variation law. The results indicate that the theoretical method is reliable, considering its good agreement with the numerical solutions. The buried depth of the shield means that the upper part of the pile is less affected by the additional thrust, leading to less deformation at the top. We recommend using a smaller shield thrust and outer diameter to control the pile's end and top displacement.

**Keywords:** shield excavation; pile foundation deformation; lateral displacement; Kerr foundation model

## 1. Introduction

In the 21st century, there has been a rise in tunnel and underground engineering in China. Cities are experiencing a surge in the planning and construction of underground spaces, which are becoming increasingly complex due to the high density of buildings. This has led to reinforced concrete pile foundations that are at risk from shield tunneling. If these foundations experience significant displacement or inclination issues during construction, these can cause further instability or cracking in the superstructure, posing a threat to the surrounding residents' lives and properties and resulting in severe losses. Accurately predicting the deformation characteristics of pile foundations and implementing protective measures during the shield tunneling process are crucial for safe construction.

Currently, the typical approach for predicting the deformation of nearby pile foundations resulting from shield construction is a two-stage analysis method. Initially, the free displacement or stress field of the soil surrounding the pile, caused by the shield construction, is determined by utilizing the Mindlin elastic solution [1]. Subsequently, soil displacement or stress is applied to the simplified pile–soil foundation model, resulting in the solution of the pile's deformation or internal force. In the initial phase, Wei et al. [2] utilized the Mindlin solution to derive a formula for surface deformation, resulting from the added thrust of the shield, shield shell friction, and soil loss. The research showed that the friction between the shield shell and the soil is relatively stable, while the positive additional thrust and soil loss are related to the construction site control and are prone to fluctuations. Feng et al. [3] simplified the tunnel into an infinite Euler–Bernoulli beam placed in a three-parameter Kerr foundation model. The research showed that the increase in the foundation modulus and tunnel depth will cause a decrease in the longitudinal

displacement and internal force of the tunnel. The increase in the tunnel's stiffness will cause a decrease in the tunnel's longitudinal displacement but an increase in the tunnel's internal force. Building upon Mindlin's solution, Liang et al. [4] considered the additional pressure on the incision front caused by soil squeezing from the cutter head, the side friction resistance of the shield shell with softening characteristics, the uneven distribution of the soft soil layer, and ground displacement resulting from synchronous grouting pressure. By combining the ground displacement caused by soil loss, they obtained the vertical and horizontal displacements of the surface and deep soil during shield construction. The research showed that the soil in front of the shield is uplifted under the action of the shield's construction. The shape is basically close to the normal distribution curve; the deep soil is squeezed and placed far away during the shield's construction. The tunnel axis moves, and its maximum horizontal displacement is near the shield axis. Wu et al. [5] deduced the ground deformation arising from uneven additional pressure on the excavation face, influenced by the slurry's density, the shield shell's friction under uneven distribution, the shield tail grouting pressure under circumferential dissipation, and the ground deformation from ground loss. They obtained the calculation formula for ground deformation resulting from a large-diameter slurry balance shield construction. The Winkler [6], Pasternak [7], and Kerr foundation models are commonly used in the subsequent phase [8]. The Kerr foundation model considers the soil's shear characteristics based on the former two foundation models and adds a spring layer, which better reflects the foundation's continuous deformation characteristics. Several studies [9–11] have utilized the Kerr foundation model to determine the deformation of infinitely long beams caused by soil excavation, such as pipelines and tunnels. However, only a handful of studies [12,13] have utilized this model to predict the pile foundation deformation caused by shield excavation. Unlike infinite long beams, pile foundations are limited in length, and the soil displacement field typically acts horizontally rather than vertically. Furthermore, most of the existing research examines tunnels crossing the side of a pile foundation [14,15], where horizontal displacement occurs on the shield excavation surface. However, a few studies have explored the lateral deformation characteristics of pile foundations located in front of the shield construction, where displacement occurs in the shield tunneling direction. As a result, further research is necessary.

We assumed that the pile foundation behaves like a vertical elastic beam with free ends. Using the Mindlin elastic solution, we derived an analytical formula for the lateral displacement of the soil in front of the shield construction. To solve the problem of the lateral displacement of the pile foundation, we introduced the Kerr foundation model. We compared the lateral displacement solution of the pile foundation in front of the shield construction with the PLAXIS 3D numerical calculation results. Using the theoretical calculation method, we further studied the influence of various factors on the displacement of the pile foundation in front of the shield construction. This provided valuable suggestions for the design and construction of comparable projects.

## 2. Lateral Displacement of Front Pile Caused by Using a Shield Machine

Figure 1 displays the excavation mechanical model of the large-diameter slurry shield machine within the stratum. During shield tunneling, the lateral displacement of the pile in front of the excavation face can be attributed to two primary factors: the additional thrust $q$ exerted by the shield perpendicular to the excavation face, and the friction $f$ created by the shield's horizontal distribution along the shield's surface.

To calculate the lateral displacement of the pile in front of the shield caused by the shield's additional thrust and friction, the Mindlin solution can be applied to determine the soil displacement resulting from these forces [1]. From there, the Kerr foundation model can be utilized to calculate the displacement of the pile in response to the soil displacement.

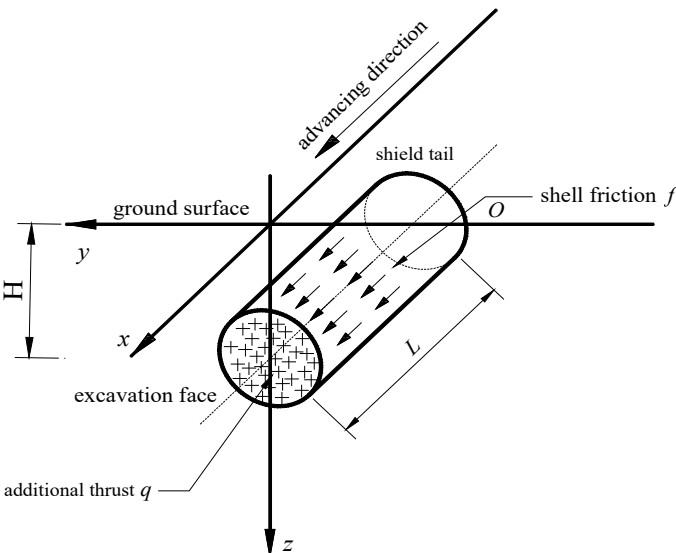

**Figure 1.** Mechanical model of large-diameter slurry shield tunneling.

### 2.1. Horizontal Displacement of Soil Caused by Shield Tunneling

The Mindlin solution is a useful tool for calculating the stress and displacement fields in a semi-infinite elastic body caused by vertical or horizontal concentrated forces in a semi-infinite space. Since the additional thrust $q$ of the shield and the horizontal distribution of the shield friction $f$ along the shield shell surface are the uniform forces distributed on the incision surface and the shield shell surface, respectively, it is necessary to perform double integral calculations and appropriate coordinate transformation on them.

The micro-element area of the circular section of the additional thrust of the shield is taken as $A = rd\theta dr$, as shown in Figure 2, and the coordinate transformation is as follows:

$$u = \begin{cases} y' = y - r\cos\theta \\ x' = x \\ z' = z \\ c = H - r\sin\theta \end{cases} \tag{1}$$

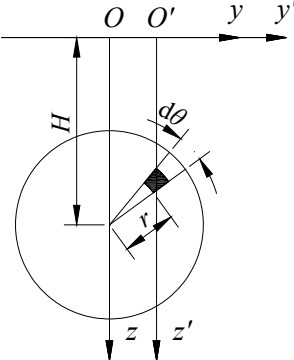

**Figure 2.** Schematic diagram of the additional thrust integral of the shield.

By substituting it into the Mindlin solution and integrating it, the displacement formula of the soil in the $x$-direction at any point in space caused by the additional thrust of the shield can be obtained:

$$u_h^q = \int_0^R \int_0^{2\pi} \frac{qr(1+\mu)d\theta dr}{8\pi E(1-\mu)} \left\{ \frac{(3-4\mu)}{R_1} + \frac{1}{R_2} + \frac{x^2}{R_1^3} \right.$$
$$+ \frac{(3-4\mu)x^2}{R_2^3} + \frac{2z(H-r\sin\theta)}{R_2^3} \left(1 - \frac{3x^2}{R_2^2}\right)$$
$$\left. + \frac{4(1-\mu)(1-2\mu)}{R_2+z+H-r\sin\theta}\left[1 - \frac{x^2}{R_2(R_2+z+H-r\sin\theta)}\right]\right\} \tag{2}$$

In the formula, $q$ represents the shield machine face thrust; $R$ represents the radius of the tunnel; $H$ represents the buried depth of the tunnel; and $R_1$ and $R_2$ are, respectively, denoted by:

$$R_1 = \sqrt{x^2 + (y - r\cos\theta)^2 + (z - H + r\sin\theta)^2} \tag{3}$$

$$R_2 = \sqrt{x^2 + (y - r\cos\theta)^2 + (z + H - r\sin\theta)^2} \tag{4}$$

The micro-element area of the cylindrical surface of the friction force acting on the shield shell is taken as $dA = Rd\theta dl$, as shown in Figure 3, and the coordinate transformation is as follows:

$$u = \begin{cases} y' = y - r\cos\theta \\ x' = x - l \\ z' = z \\ c = H - R\sin\theta \end{cases} \tag{5}$$

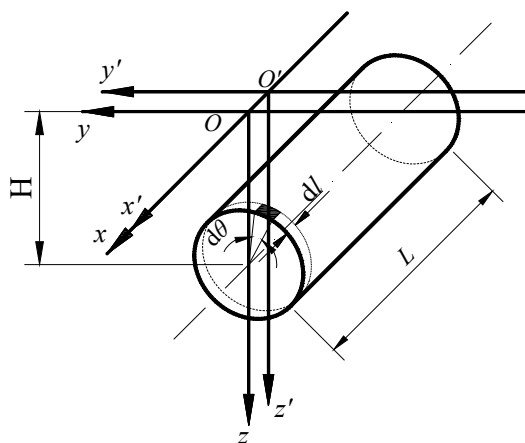

**Figure 3.** Thrust integral diagram of shield machine.

By substituting it into the Mindlin solution and integrating it, the displacement formula of soil in the $x$-direction at any point in space caused by the shield shell friction can be obtained:

$$u_h^f = \int_0^L \int_0^{2\pi} \frac{fRd\theta dl}{16\pi G(1-\mu)} \left\{ \frac{(3-4\mu)}{R_1} + \frac{1}{R_2} + \frac{(x-l)^2}{R_1^3} + \frac{(3-4\mu)(x-l)^2}{R_2^3} + \frac{2(H-R\sin\theta)z}{R_2^3}\left(1 - \frac{3(x-l)^2}{R_1^2}\right) + \right.$$
$$\left. \frac{4(1-\mu)(1-2\mu)}{R_2+z+H-R\sin\theta}\left[1 - \frac{(x-l)^2}{R_2(R_2+z+H-R\sin\theta)}\right]\right\} \tag{6}$$

In the formula, $L$ is the shield shell length; $R_1$ and $R_2$ are, respectively, denoted by:

$$R_1 = \sqrt{(x-l)^2 + (y - R\cos\theta)^2 + (z - H + R\sin\theta)^2} \tag{7}$$

$$R_2 = \sqrt{(x-l)^2 + (y - R\cos\theta)^2 + (z + H - R\sin\theta)^2} \tag{8}$$

The friction force $f$ acting between the shield shell and the surrounding soil at any angle $\theta$ on the shield shell can be calculated using the formula for the shield shell friction,

which considers soil softening and unevenness. This formula is developed by referring to Liang Rong's method [4].

$$f = \tau = \beta_s \sigma'_\theta \tan \delta' \tag{9}$$

In the formula, $\tau$ represents the shear stress that occurs between the shield shell and the nearby soil; $\beta_s$ represents the ratio of the residual friction resistance to the ultimate friction resistance, and it ranges from 0.83 to 0.97. $\sigma'_\theta$ is the radial normal stress of the soil at any point on the shield; $\delta\prime$ is the interface friction angle between the shield shell and the surrounding soil, which can be obtained by referring to the results of Potyondy's [16] interface shear test.

### 2.2. Pile Foundation Displacement Caused by Soil Displacement

The additional load $p$ acting on the pile foundation can be approximately expressed by the displacement field of the soil.

$$p = \eta \left[ \frac{mk}{m+1} S_x(z) - \frac{m^2 G}{(m+1)^2} S''_x(z) \right] \tag{10}$$

In the formula, $\eta$ represents the load correction coefficient, and it ranges from 0.4 to 1.0; $S_x(z)$ represents the lateral displacement of the soil free field caused by the tunnel's construction.

The equilibrium differential equation of a single pile on the Kerr foundation is:

$$-\frac{EIG}{Dc} \frac{d^6 w_l}{dz^6} + \frac{EI(c+k)}{Dc} \frac{d^4 w_l}{dz^4} - G \frac{d^4 w_l}{dz^2} + k w_l = p \tag{11}$$

In the formula, $w_l$ represents the lateral displacement of the pile foundations; $z$ represents the buried depth of the pile foundations; and $c$ represents the spring stiffness on the right side of the shear layer. Assuming $c = mk$, $m$ represents the second layer spring parameter of the foundations; Feng et al. [17] used the finite difference method to solve Equation (11), which is a sixth-order differential equation, combined with the boundary conditions of the pile top and the pile end to obtain the horizontal (*x*-direction) displacement equation of the pile foundations:

$$\{W_l\} = [K]^{-1} \cdot \{P\} \tag{12}$$

$$\{W\} = \{w_0, w_1, ..., w_{n-1}, w_n\} \tag{13}$$

$$\{P\} = \{p_0, p_1, ..., p_{n-1}, p_n\} \tag{14}$$

The stiffness matrix [*K*] of the soil in the horizontal (lateral) direction when the top and end of the pile are free:

$$[K] = \begin{bmatrix}
\delta+2\gamma+4\beta-8\alpha & -4\beta-10\alpha & 2\beta+2\alpha & 2\alpha \\
\gamma+2\beta+6\alpha & \delta-\beta-2\alpha & \gamma-\alpha & \beta & \alpha \\
\beta-2\alpha & \gamma-\alpha & \delta & \gamma & \beta & \alpha \\
\alpha & \beta & \gamma & \delta & \gamma & \beta & \alpha \\
& \ddots & & \ddots & \ddots & \ddots & \ddots & \ddots \\
& & \alpha & \beta & \gamma & \delta & \gamma & \beta & \alpha \\
& & & \alpha & \beta & \gamma & \delta & \gamma-\alpha & \beta-2\alpha \\
& & & & \alpha & \beta & \gamma-\alpha & \delta-\beta-2\alpha & \gamma+2\beta+6\alpha \\
& & & & & 2\alpha & 2\beta+2\alpha & -4\beta-10\alpha & \delta+2\gamma+4\beta-8\alpha
\end{bmatrix} \tag{15}$$

$$\begin{cases} \alpha = -\frac{EIG}{Dch^6} \\ \beta = \frac{6EIG}{Dch^6} + \frac{EI(c+k)}{Dch^4} \\ \gamma = -\frac{15EIG}{Dch^6} - 4\frac{EI(c+k)}{Dch^4} - \frac{G}{h^2} \\ \delta = \frac{20EIG}{Dch^4} + 6\frac{EI(c+k)}{Dch^4} + 2\frac{G}{h^2} + k \end{cases} \tag{16}$$

The expressions of each item in the formula are substituted into the equation, and the matrix and the expressions are expressed using MATLAB software (https://www.mathworks.com/products/matlab.html). The lateral displacement of the pile foundation caused by the soil displacement can be solved.

### 3. Numerical Simulation Verification Results

To ensure the validity of the theoretical calculations, the numerical simulation is utilized in conjunction with the ongoing project to compare their results. As depicted in Figure 4, the project consists of a single C40 concrete cast-in-place pile with a diameter of 0.9 m in the axial direction of the shield tunnel. The shield tunnels towards the single pile, with the constant additional thrust *p* and friction *f* of the shield shell throughout the process. The calculation assumes a horizontally uniform soil distribution, without accounting for the influence of the groundwater level. For ease of verification, both the theoretical calculations and the numerical simulation simplify the soil into a homogeneous layer. According to the geological survey report on the project, the soil parameters are detailed in Table 1.

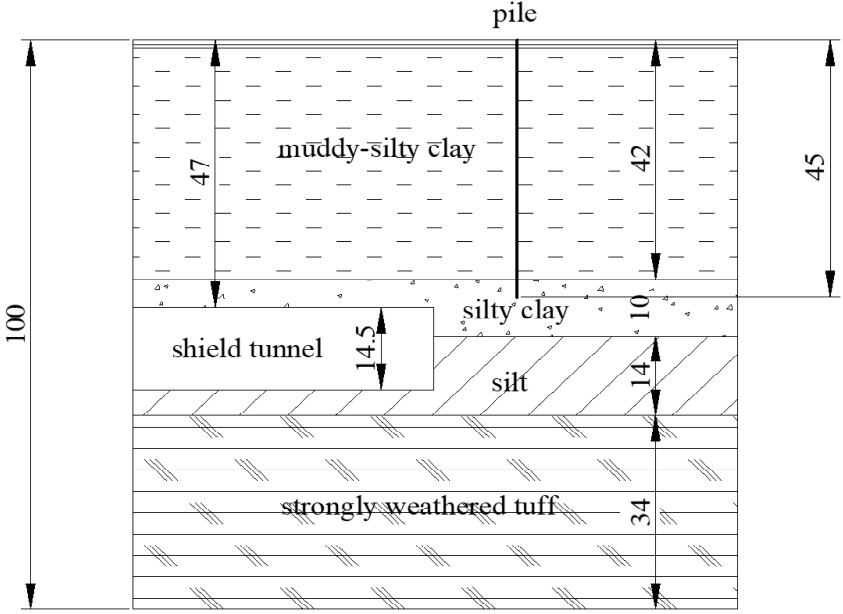

**Figure 4.** Schematic diagram of the engineering case (unit: m).

**Table 1.** Parameters of soil and rock.

| Soil Layer | Unit Weight $\gamma$/kN·m$^{-3}$ | Elastic Modulus $E_s$/MPa | Poisson Ratio $\nu$ | Cohesion Force $C$/kPa | Internal Frictional Angle $\psi$/° |
|---|---|---|---|---|---|
| silty clay | 19 | 3.8 | 0.35 | 17 | 10.3 |

The numerical model of the influence of the shield construction on the displacement of the front pile is established using PLAXIS three-dimensional finite element software (https://www.bentley.com/software/plaxis-3d/). The constitutive model of the soil adopts the simple, fast, and easy-to-converge Mohr–Coulomb elastic–plastic model, which is often used in engineering design. The shield shell is simulated using the plate element.

In order to avoid the influence of the boundary conditions of the model, the size of the model should be at least three times larger than the size of the main structure. The model is 200 m long, 100 m wide, and 120 m high. The upper surface of the model is free, the horizontal constraint is applied to the side, and the bottom surface is completely fixed. The rationality verification of this numerical model based on the actual field data is referred to [18]. The grid sensitivity is checked at the time of the model's building to ensure that the numerical model is consistent with the actual engineering situation. In this model, the soil element uses a 10-node tetrahedral element, the plate element uses a 6-node triangular element, and the 12-node interface element is used to simulate the interaction between the structure and the soil. The global density of the model is set to medium, and a total of 235,051 units are generated. The specific material parameters are shown in Table 2. The numerical model is shown in Figure 5.

**Table 2.** Input parameters of the plate element.

| Structure | Material | Equivalent Thickness $D$/m | Unit Weight $\gamma$/kN·m$^{-3}$ | Equivalent Elastic Modulus $E$/kPa | Poisson Ratio $\nu$ |
|---|---|---|---|---|---|
| shield shell | C60 | 0.6 | 12 | $3.60 \times 10^{7}$ | 0.2 |

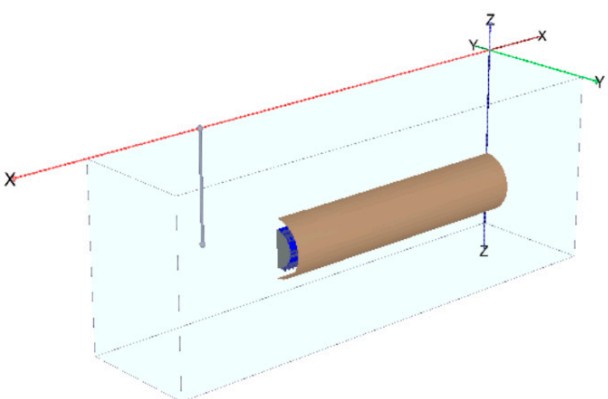

**Figure 5.** PLAXIS 3D model with pile.

MATLAB software is used to assist in the theoretical calculation. First, the displacement of the soil during the shielding process is obtained using the Mindlin solution. Then, the lateral displacement of the pile foundation under the soil displacement is solved by combining it with the horizontal displacement equation of the pile foundation under the Kerr foundation model. The calculation ignores the upper boundary conditions, and the buried depth of the shield is determined from the height of the pile top. The calculation parameters of the theoretical model are presented in Table 3.

In Figure 6, we can see the lateral displacement of the pile obtained from both the numerical simulation and the theoretical calculation. The two methods show similar trends in the displacement of the pile foundation. However, there are some differences in their results. For the upper half of the pile, the numerical calculation produces slightly larger displacement results than the theoretical calculation does. The difference between the two methods is only 0.47 mm when the distance between the shield surface and the pile is 10 m. On the other hand, for the lower part of the pile, the numerical solution gives smaller displacement results than the theoretical solution does. The difference between the two increases as the distance between the excavation surfaces decreases. At 10 m between the excavation surfaces, the difference in the displacement results at the pile end is 2.1 mm, which represents an error of about 19.85%.

**Table 3.** Parameters of theoretical calculation.

| Design Conditions | Values |
|---|---|
| Shield length $L$/m | 12 |
| Shield radius $R$/m | 7.25 |
| Additive thrust $p$/kPa | 20 |
| Friction resistance ratio $\beta$ | 0.9 |
| Interface friction angle $\delta'$/° | 8.0 |
| Coefficient of earth pressure at rest $K_0$ | 0.5 |
| Foundation parameters of Kerr model $m$ | 4 |

**Figure 6.** Comparison of calculated and theoretical values of lateral displacement of pile foundations. (**a**) x = 10 m; (**b**) x = 20 m.

Overall, the theoretical calculation method provides results that are close to those of the numerical simulation. The theoretical calculation method is better suited for calculating the displacement of the pile in front of the shield center line under the action of the shield's force, and the displacement result of the pile end obtained using the theoretical calculation is larger than that of the numerical solution, making it safer for construction. Additionally, using MATLAB programming language for calculation purposes can reduce the amount of engineering required compared to that needed for the numerical simulation.

## 4. The Influence of Shield Construction on the Displacement of the Front Pile Foundation Discussion

It is assumed that there is a single pile in front of the tunnel, and both the soil and the pile are linear, elastic, and isotropic. The pile diameter is 0.9 m, the pile length is 45 m, the elastic modulus is 25 GPa, the soil elastic modulus is 8 MPa, and the Poisson's ratio is 0.5. MATLAB programming language is used to calculate the lateral displacement of the pile under the combined action of the additional thrust of the shield and the friction of the shield shell in the shield construction, and the lateral displacement of the pile foundation under different additional thrusts, excavation surface distances, and tunnel diameters is analyzed. The additional thrusts are 20 kPa, 50 kPa, 100 kPa, and 150 kPa, respectively. The distances between the excavation faces are 1, 1.5, 2.0, 2.5, and 3.0*D*, respectively (*D* is the shield diameter). The tunnel diameters are 11.5 m, 12.5 m, 13.5 m, 14.5 m, and 15.5 m.

In Figure 7, the impact of the distance between the pile foundation and the excavation face on the lateral displacement of the pile foundation is demonstrated under varying additional thrusts of the shield. The shield's outer diameter is 14.5 m, and due to its significant depth, the excavation surface's proximity has only a small effect on the pile top's lateral displacement, but greatly influences the lower part of the pile body, resulting in the inclination of the pile foundation. Therefore, when constructing the shield near the pile foundation, the evaluation standard of the safety of the pile foundation should consider not only its displacement, but also its inclination.

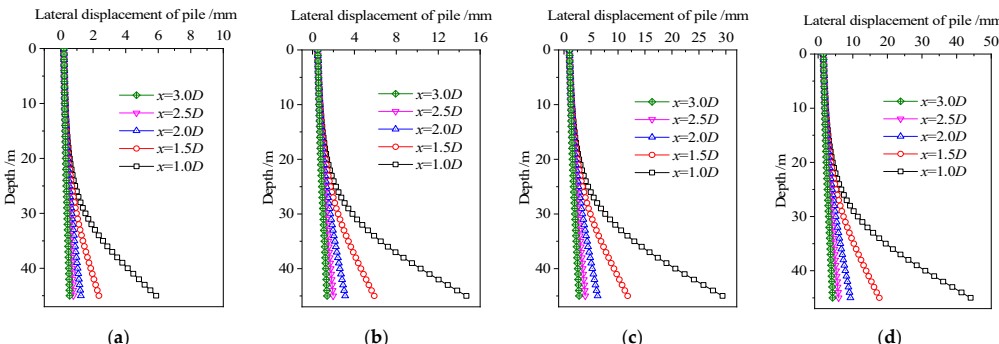

**Figure 7.** The lateral displacement of the pile foundation under the influence of the excavation surface distance and additional thrust of the shield. (**a**) $q$ = 20 kPa, (**b**) $q$ = 50 kPa, (**c**) $q$ = 100 kPa, and (**d**) $q$ = 150 kPa.

As the additional thrust of the shield increases, the lateral displacement of the pile foundation gradually increases as well. The decrease in the excavation face distance leads to an exponential increase in the lateral displacement of the pile foundation. When the additional shield thrust is maintained within a reasonable range of ±50 kPa, the maximum impact on the lateral displacement of the pile foundation can be controlled within 20 mm. However, at 150 kPa, the disturbance caused by the additional thrust of the shield is significant. When the distance from the pile foundation is equal to the outer diameter, the maximum lateral displacement of the pile foundation is almost 45 mm or 1% of the pile length, which jeopardizes the pile foundation's upper structure's safety. Thus, during the construction of the shield near the pile foundation, especially when the excavation surface's proximity is minimal, minimizing the shield's thrust is critical to reduce its impact on the pile foundation displacement.

Figure 8 displays how the diameter of a shield affects the lateral displacement of a pile foundation when the excavation face is 29 m away, or two times larger than the shield diameter. The deformation pattern of the foundation is caused by the varying shield diameter, excavation face distance, and additional shield thrust, which all have similar effects. While increasing the shield diameter leads to greater soil displacement, it has a smaller impact compared to those of the shield thrust and excavation face distance. Therefore, controlling the shield thrust is crucial in managing stratum deformation during construction, and the tunnel-to-pile-foundation distance should be determined appropriately during the design process. Unlike changing the excavation face distance, altering the shield diameter has a more significant effect on the pile top displacement, indicating that different shield diameters cause larger lateral displacement changes on the surface.

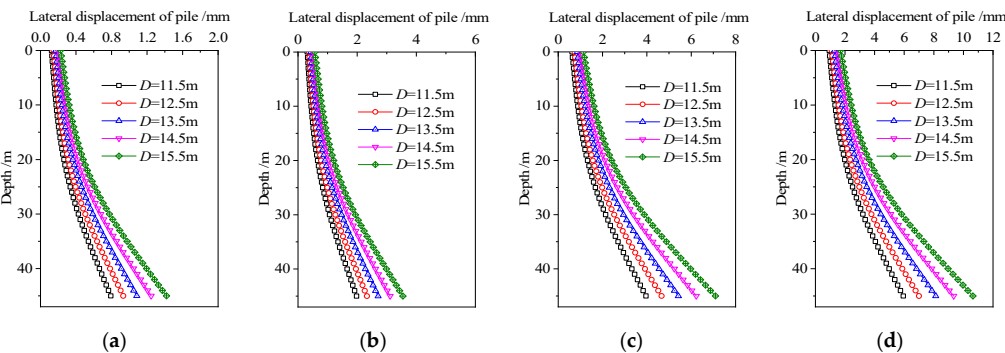

**Figure 8.** The lateral displacement of the pile foundation under the influence of the shield's outer diameter. (**a**) $q$ = 20 kPa, (**b**) $q$ = 50 kPa, (**c**) $q$ = 100 kPa, and (**d**) $q$ = 150 kPa.

## 5. Conclusions

In this paper, by aiming to understand the influence of shield construction on the displacement of a pile foundation, a mechanical model of the influence of shield construc-

tion on the front pile foundation is established. The displacement of soil during shield construction is obtained using the Mindlin solution. Combined with the horizontal displacement equation of the pile foundation using the Kerr foundation model, the lateral displacement expression of the pile foundation under soil displacement is obtained. Then, through the actual project as a verification example, the finite element model is established by using PLAXIS three-dimensional finite element software, and the theoretical solution expression is solved with MATLAB. The reliability of the theoretical solution is verified by comparing the theoretical and numerical solutions. Finally, the lateral displacement of the pile foundation caused by large-diameter shield tunneling is analyzed using the theoretical solution, and the deformation law of the pile foundation caused by the additional thrust of the shield, excavation distance, and change in the outer diameter of the shield is explored. The conclusions are as follows:

(1)  The Mindlin elastic solution and simplified pile–soil foundation model can be used to solve the lateral displacement of the pile foundation caused by the deep-buried shield, which can accurately predict the overall lateral displacement of the pile foundation in front of the shield. MATLAB programming language can be used to realize a simple and efficient calculation, which can provide fast and accurate guidance for shield design and construction.

(2)  Comparing the numerical and theoretical results of the influence of the shield construction on the displacement of the front pile foundation, for the upper part of the pile foundation, the theoretical calculation is similar to the numerical one. For the lower part of the pile foundation, when the shield is closer to the pile foundation, the theoretical calculation value is larger, and the result is conservative.

(3)  A larger shield additional thrust and closer excavation face distance will cause larger displacement in the lower part of the pile foundation. In the design, the distance between the excavation face and the pile foundation should be larger than the pile diameter. During construction, when the distance between the excavation face and the pile foundation is small, the shield thrust should be reduced as much as possible to reduce the influence on the pile tip displacement and the pile foundation inclination.

(4)  Compared with changing the additional thrust of the shield and the distance from the excavation face, changing the outer diameter of the shield has only a small effect on the overall lateral displacement of the pile foundation. In order to control the deformation of the stratum, it is suggested that the distance between the tunnel and the pile foundation should be reasonably determined in the design stage, and the thrust of the shield should be reasonably controlled during the construction. Changing the outer diameter of the shield has a large influence on the displacement of the pile top. The key to controlling the displacement of the pile top and the ground surface is the outer diameter of the shield. It is suggested that the influence of the outer diameter of the shield should be considered when calculating the displacement of the pile top and the ground surface in the design stage.

**Author Contributions:** Conceptualization, Y.S. and F.W.; methodology, F.W.; software, C.W.; validation, Z.M. and Y.S.; formal analysis, F.W.; investigation, F.W.; resources, Y.S.; data curation, Y.S.; writing—original draft preparation, F.W.; writing—review and editing, F.W.; visualization, F.W.; supervision, Y.S.; project administration, Y.S.; funding acquisition, Y.S. All authors have read and agreed to the published version of the manuscript.

**Funding:** This research was supported by the Science and Technology Project of Jiangsu Provincial Transportation Department (grant number 2022Y13) and the Science and Technology Plan Project of CSCEC 7th Division (grant number CSCEC7b-2022-Z-6).

**Institutional Review Board Statement:** Not applicable.

**Informed Consent Statement:** Not applicable.

**Data Availability Statement:** Data are contained within the article.

**Conflicts of Interest:** The authors declare no conflict of interest.

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
