# Peer review of "Analysis of Lateral Displacement of Pile Foundation Caused by Large-Diameter Shield Tunneling"

_applsci, doi:10.3390/app14010272_

Round 1
Reviewer 1 Report
Comments and Suggestions for Authors
In the present article, the authors have tried to analyze the lateral displacement of the pile foundation caused by the large-diameter shield tunneling. Please find below review comments which may help the authors to improve the quality of the manuscript.
1. As mentioned in the article the Mindlin elastic solution has been used to determine the lateral displacement. What will happen if the soil yields and becomes plastic?
2. Torsional loads may occur due to the eccentricity of the applied lateral loads. Are these considered in the analysis?
3. Why the Mohr-Coulomb elastoplastic model has been used in the present study. Suggest a discussion.
4. Add references for parameters in Tables 1 and 2.
5. How the flow potential of the yield surface is computed using the Plaxis 3D computational software?
6. How does the dilation of the soil affect the behavior of the present numerical model?
7. As the Plaxis 3D computational analysis is an important part of the study, it is recommended to explain all the minor details of the analysis.
8. What are the boundary conditions applied in the present numerical analysis for specified loading conditions?
9. Mention details of mesh size, the shape of the finite element, the number of elements employed, etc. in the research article as this is an important part of the study.
10. Was a sensitivity analysis performed to confirm that model convergence was achieved with the selected mesh size?
11. How the initial stress field has been employed in the present numerical model?
12. Incorporate a step-by-step example (Mindlin solution calculation, Matlab used in the present study, etc.) in the article, which may help the young researchers follow your research.
13. Add more pictures of the numerical analysis model including geometry, mesh, results, etc. Improve Figure 5.
14. The conclusion part is not written well. Suggest improving.
Comments on the Quality of English LanguageIt can be improved.
Reviewer 2 Report
Comments and Suggestions for Authors
The similarity is 35% with bibliography and 31% excluded it. It considered a hight percentage from the reviewer's perspective. The authors should modify their article to reduce this percentage.
Comments on the Quality of English LanguageI strongly advise the publication of this paper after some minor review: - Minor editing of English language required especially in the introduction.
Reviewer 3 Report
Comments and Suggestions for Authors
This paper proposed a theoretical method to analyze Lateral Displacement of Pile Foundation Caused by 2 Large Diameter Shield Tunneling. The work is of scientific significance. However, the presentation and the English language should be improved before it could be accepted.
1) The English language should be extensively checked and improved.
2) The numerical method to solve the differential equation of the displacement should be introduced in detail.
3) The Plaxis model used for validation should be introduced in detail. And the accuracy of the model should be verified.
4) Section 3 and 4 simply introduce the result, however, further discussion is suggested. For example, the reason of conclusion point 2 should be explained in further discussion.
Comments on the Quality of English LanguageEnglish writting has a lot of grammar mistakes. Extensive editing of English language is required.
Reviewer 4 Report
Comments and Suggestions for Authors
the paper is so good and have a valuable geotechnical knowledge in the filed of tunnel installtion
But the author does not consider improtant thing or neglecting the exsitance of shaft of caisson ( room of installtion and their induced larg horizontal force in sourrounding media)
Comments on the Quality of English Languagemay be imporved by native english and check of plagrism
Reviewer 5 Report
Comments and Suggestions for Authors
Dear Editor/Authors
I have assessed the manuscript “Analysis of Lateral Displacement of Pile Foundation Caused by Large Diameter Shield Tunneling” article.
Since the content and discussion part of the study is weak, I do not find it appropriate to publish this study as an article. The manuscript is not suitable for publishing in its current state.
I wish success to the author(s).
Best regards
Comments on the Quality of English LanguageDear Editor/Authors
I have assessed the manuscript “Analysis of Lateral Displacement of Pile Foundation Caused by Large Diameter Shield Tunneling” article.
Since the content and discussion part of the study is weak, I do not find it appropriate to publish this study as an article. The manuscript is not suitable for publishing in its current state.
I wish success to the author(s).
Best regards
Round 2
Reviewer 1 Report
Comments and Suggestions for Authors
The contents of the study are not suitable to be published as a research article.
Comments on the Quality of English LanguageIt can be improved
Author Response
Thank you for your comments and comments.
Reviewer 3 Report
Comments and Suggestions for Authors
The questions I proposed are not properly addressed. Neither the verification of the Plaxis model, nor further discussion is properly added.
As to the verification of the Plaxis model, it better be verified against tests. And mesh sensitivity check and other issues should be clearly introduced.
Comments on the Quality of English LanguageMinor editing of English language required
Author Response
The questions I proposed are not properly addressed. Neither the verification of the Plaxis model, nor further discussion is properly added.
As to the verification of the Plaxis model, it better be verified against tests. And mesh sensitivity check and other issues should be clearly introduced.
RESPONSE:
I agree with your opinion, but the numerical model established in this paper is a model widely applicable to engineering design in China and is used as the verification of theoretical solutions, so validation is not added in the paper.
We have revised some statements and added an reference has varified the model in the article to the following:
“In this model, the soil element uses a 10-node tetrahedral element, the plate element uses a 6-node triangular element, and the 12-node interface element is used to simulate the interaction between the structure and the soil. The global density of the model is set to medium, and a total of 235051 units are generated. In this paper, a model widely applicable to engineering practice is established, and the rationality verification of this numerical model based on the actual field data is referred to [xx]. The grid sensitivity is checked at the time of model building to ensure that the numerical model is consistent with the actual engineering situation.
[xx.LI Mingrui, CHEN Guoping, FAN Xiujiang, XU Pingyuan, DING Shilong, SUN Zhihao, XU Changjie.Numerical study and parametric analysis of influence of tunnel excavation on adjacent pile foundation[J].JOURNAL OF CIVIL AND ENVIRONMENTAL ENGINEERING,2022,44(1):45-52.10.11835/j.issn.2096-6717.2021.034”
Reviewer 4 Report
Comments and Suggestions for Authors
the paper is technically so good but the author may consider the surface ground loss of settlement of trough that placed the pile
Also study the vertical deformation of pile adjacent to tunniling process
Author Response
Thank you for your comments and comments.
Have a nice day!
Reviewer 5 Report
Comments and Suggestions for Authors
Dear Editor/Authors
I reassessed the article “Analysis of Lateral Displacement of Pile Foundation Caused by Large Diameter Shield Tunneling” after the corrections were made. Since the study has been improved in terms of content, discussion and language. I think this submitted article lacks from the below issues:
* What is the difference between this study from other studies in the literature. Because there are lots of similar studies in the literature and please make a comparison. Write the similar/non-similar points and emphasize your originality.
* The introduction and literature review section should be improved. Bring along as much as you can the baseline scientific proof for the knowledge you are going to explore in further sections. The literature is better to be updated based on the most recent works.
* The article should be enriched with figures.
I believe that a more understandable article will be obtained when all these points are evaluated and added to the study.
I think that the study can be published in the journal after these revisions.
I wish success to the authors in their study.
Best regards,
Author Response
Dear Reviewer,
Thank you for your valuable feedback and thorough review.
1、* What is the difference between this study from other studies in the literature. Because there are lots of similar studies in the literature and please make a comparison. Write the similar/non-similar points and emphasize your originality.
RESPONSE:
I agree with you that we will add the following to the article:
“However, only a handful of studies [12,13] have utilized this model to predict the pile foundation deformation caused by shield excavation. Unlike infinite long beams, pile foundations are limited in length, and the soil displacement field typically acts horizontally rather than vertically. Furthermore, most of the existing research examines tunnels crossing the side of a pile foundation [14,15], where horizontal displacement occurs on the shield excavation surface. However, a few studies have explored the lateral deformation characteristics of pile foundations located in front of the shield construction, where displacement occurs in the shield tunneling direction. As a result, further research is necessary.”
2、* The introduction and literature review section should be improved. Bring along as much as you can the baseline scientific proof for the knowledge you are going to explore in further sections. The literature is better to be updated based on the most recent works.
RESPONSE:
I agree with you that we will add the following to the article:
“Subsequently, soil displacement or stress is applied to the simplified pile–soil foundation model, resulting in the solution of the pile's deformation or internal force. In the initial phase, Wei et al. [2,3] utilized the Mindlin solution to derive a formula for surface deformation, resulting from the added thrust of the shield, shield shell friction, and soil loss. The research showed that the friction between the shield shell and the soil is relatively stable, while the positive additional thrust and soil loss are related to the construction site control and are prone to fluctuations. Building upon Mindlin's solution, Liang et al. [4] considered the additional pressure on the incision front caused by soil squeezing from the cutter head, the side friction resistance of the shield shell with softening characteristics, the uneven distribution of the soft soil layer, and ground displacement resulting from synchronous grouting pressure. By combining the ground displacement caused by soil loss, they obtained the vertical and horizontal displacements of the surface and deep soil during shield construction. The research showed that the soil in front of the shield is uplifted under the action of shield construction. The shape is basically close to the normal distribution curve; the deep soil is squeezed and far away during the shield construction. The tunnel axis moves, and its maximum horizontal displacement is near the shield axis.
And we will add an recent work in the paper:
“Feng et al. simplified the tunnel into an infinite Euler-Bernoulli beam placed in a three-parameter Kerr foundation model. The research showed that the increase of foundation modulus and tunnel depth will cause the decrease of longitudinal displacement and internal force of the tunnel. The increase of tunnel stiffness will cause the decrease of tunnel longitudinal displacement but the increase of tunnel internal force.”
3、* The article should be enriched with figures.
RESPONSE:
I agree with you, we will increase the detailed figures in some parts of the article, modified like the following:
“When the additional shield thrust is maintained within a reasonable range of ± 50 kPa, the maximum impact on the lateral displacement of the pile foundation can be controlled within 20 mm. However, at 150 kPa, the disturbance caused by the additional thrust of the shield is significant. When the distance from the pile foundation is equal to the outer diameter, the maximum lateral displacement of the pile foundation is almost 45 mm or 1% of the pile length, which jeopardizes the pile foundation's upper structure's safety. Thus, during the construction of the shield near the pile foundation, especially when the excavation surface's proximity is minimal, minimizing the shield's thrust is critical to reduce its impact on the pile foundation displacement.”
Thank you once again for your guidance and assistance. We look forward to your feedback.
Best regards,
Round 3
Reviewer 1 Report
Comments and Suggestions for Authors
The authors have incorporated the review comments. The article may be accepted.
Comments on the Quality of English LanguageIt can be improved.